# Philosophy and ethics of de-extinction

Jay Odenbaugh 

Department of Philosophy, Lewis & Clark College, Portland, OR, USA

## Overview Review

Ethics and policy; Ancient DNA; Biodiversity conservation; Conservation; Deextinction

**Author for correspondence:**
Jay Odenbaugh,
Email: jay@lclark.edu

## Abstract

In this essay, we explore the philosophical and ethical issues concerning de-extinction. First, we will characterize what de-extinction is. This requires clarification of the process of extinction. Second, we consider whether de-extinction is even possible. There are a variety of arguments involving the nature of species that purport to show that once they have disappeared they cannot be resurrected. Third, we examine whether de-extinction is morally permissible. There are arguments that suggest we are obligated to do it based on restorative justice and biodiversity conservation. There are other arguments that conclude we are not permitted to do so based on considerations of animal welfare, hubris and the allocation of conservation resources.

## Impact statement

De-extinction is a conceptually and ethically challenging topic. Through careful philosophical reflection, we can recognize it is a logically coherent course of action, though it is not always ethically advisable. Sometimes it is morally impermissible.

## What is de-extinction?

Many think that once a species goes extinct, the species ceases to exist. However, recent scientific work on de-extinction suggests the possibility of "resurrecting" lost species. From a philosophical perspective, this raises two questions. First, is de-extinction even possible? Second, should it be done even if we can do it? In this essay, we consider proposed answers to both questions.

To begin, we need to get clearer on what exactly de-extinction is. One problem that immediately appears is that de-extinction seems to be impossible *by definition.* Here is an example of this worry from Alastair S. Gunn.

> To say that a species is extinct is to say something about its past as well as its present status – although there used to be moa, they no longer exist. It may be argued that *extinct* also says something about the future of a class – that once it becomes a null class, it can never come to have members again. It may even be claimed that this is what *extinct* means. If so, then the question, "Can extinct species be recreated?" is answered negatively by resort to what is sometimes called "definitional stop." (Gunn, 1991, p. 299)

If a species is extinct, then it cannot exist in the future. If it cannot exist in the future, then it cannot be resurrected. But de-extinction just is the resurrection of an extinct species. Therefore, de-extinction is impossible. Before I examine this argument, I want to consider a terminological worry. As Beth Shapiro notes, "de-extincted" is an awkward and unappealing verb. I will describe the process of de-extinction as *resurrection* when we can bring back the very species that disappeared. I will use the term *recreation* when we create organisms very similar to those of an extinct species but that are not necessarily the same species as it. Resurrection is sometimes associated with bringing back the very organism who has died, and this is *not* what is being discussed in debates regarding de-extinction. We are discussed higher levels of organization such as populations, species and the like.

To see the error in this argument, we must consider what "extinct" means. The Concise Oxford Dictionary of Ecology defines it this way, "Applied to a taxon no member of which is living at the present time" (Allaby, 1992, p. 150). However, this would imply when plants of a species are dormant, it has gone extinct. Yashina et al. (2012) found 30,000-year-old fruit tissue from narrow-leafed campion (*Silene stenophylla*) in the Siberian permafrost and were able to successfully create fertile plants. Likewise, gene banks such as the Svalbard Global Seed Vault serve as repositories for regenerating species.[1] Thus, this definition does not capture what extinction is.

A different definition comes from paleobiologist Michael Hannah. He writes, "The extinction of a species occurs when the last individual belonging to that species dies" (Hannah, 2021, p. 36). His definition does not imply that the narrow-leafed campion went extinct when there were no living members. This is because the *last* member had not died. So, the simplest response to the argument that de-extinction is impossible by definition is that in cases where the resurrection of a

species occurs, we were simply wrong that the species had gone extinct. We thought it was extinct, but it was not. This does occur occasionally in other contexts. For example, the black-browed babbler (*Malacocincla perspicillata*) had not been observed in 170 years and was presumed extinct. However, in 2020, a team of researchers found a specimen in Borneo (Akbar et al., 2020).

One might object that this response trivializes the notion of extinction (and thus the impossibility of de-extinction). For any given species, they might be resurrected and so we are never able to declare a species extinct. For example, Ben Novak suggests that we consider "reproductively competent single cells" as members of an "evolutionary torpid species" (Novak, 2018, p. 9). However, even in the best of circumstances, information from DNA cannot be extracted after approximately 1.5 million years (Allentoft et al., 2012). Thus, for any species which has disappeared for that length of time or more, we simply cannot resurrect them. In some cases, we can confidently declare a species extinct. The concept of extinction is not trivialized by de-extinction.

Let us characterize what de-extinction is. *De-extinction* is the process of creating living organisms that are similar (often genetically similar) to members of extinct species (Sandler, 2017, p. 1).[2] Some authors augment this definition with notions of ecological function. For example, Ben Novak writes,

> [D]e-extinction is the ecological replacement of an extinct species by means of purposefully adapting a living organism to serve the ecological function of the extinct species by altering phenotypes through means of various breeding techniques, including artificial selection, back-breeding and precise hybridization facilitated by genome editing. (Novak, 2018, p. 5)

These definitions of de-extinction are inclusive. The reason they are inclusive is that the techniques associated with de-extinction do not require that a species be resurrected but also may be recreated.[3] To see why, let us consider these three techniques (Shapiro, 2015, 2017). There is back-breeding, cloning through somatic cell nuclear transfer (SCNT), and genetic engineering. This also allows us to explore examples of current or future de-extinction.

Back-breeding uses selective breeding to bring back ancestral traits in extant populations of organisms.[4] This technique has been used in bringing back traits in species similar to the auroch (*Bos primigenius*) which is the ancestor of modern cattle. The last known auroch died in Poland in 1627. Fascinated with the animal, German brothers Heinz and Lutz Heck selected for cattle that had large horns, large body size, and were more aggressive than most cattle. In 1983, a group of 32 "Heck cattle" were released into a nature preserve in the Netherlands, the Oostvaardersplassen.[5] Notice that back-breeding recreates the auroch by virtue of reestablishing auroch-like traits though the auroch is not resurrected. This is because in recreation we create organisms very similar to those of an extinct species even when they are not members of the same species.

Cloning uses SCNT to create a genetic copy of an organism. The nucleus from an adult somatic cell is placed into an enucleated egg cell. The host egg cell becomes an undifferentiated pluripotent stem cell and has an identical nuclear genome sequence to the donor of the somatic cell.[6] For example, in 2003, a cloned bucardo (*Capra pyrenaica pyrenaica*) calf was born. The bucardo is an extinct subspecies of the Pyrenean ibex. This individual died soon after they were born due to a lung deformity. One of the most discussed possibilities for de-extinction is the woolly mammoth (*Mammuthus primigenius*) which went extinct in Eurasia and North America about 8,000–10,000 years ago. The last population of dwarf woolly mammoths disappeared on Siberia's Wrangel Island

4,000 years ago (Vartanyan et al., 1995; Stuart et al., 2004). Fortunately, there are many mammoth bones in the subarctic from which cells (or at least chromosomes) could be extracted. Woolly mammoths are more closely related to Asian elephants (*Elephas maximus*) than African ones (*Loxodonta*). In principle, they could be cloned with Asian elephant mothers. One serious ethical problem here is that the Asian elephant is endangered, and egg harvesting, implantation along with pregnancy would likely harm them. Of course, some surrogate species are not endangered. We will consider issues of animal welfare later.

For approximately 100,000 years, there was a large area of productive grasslands on which horses, bison, woolly rhinoceroses and woolly mammoths lived. This "mammoth steppe" disappeared 10,000 years ago (Zimov, 2005). Mammoths, and other megafauna, disappeared due to a combination of climate change and human predation. This in turn leads to the disappearance of these grasslands. These grasslands created carbon-rich materials such as dead plant roots frozen in the soil and permafrost. However, anthropogenic climate change is melting the permafrost and this will rot the plant material releasing this carbon (or methane) into the atmosphere. Sergey A. Zimov, founder of Pleistocene Park, proposes resurrecting woolly mammoths to slow down permafrost melting. Snow-covered forests have a lower albedo than do snow-covered grasslands. The recreation of snow-covered grasslands would prevent a greater amount of permafrost melt than currently existing snow-covered forests. Grazing animals would trample the snow which brings colder air to chill the permafrost (this air is often colder than the permafrost itself). Zimov and his colleagues have established herds of herbivores including horses, moose, reindeer, muskox and yak. Unlike woolly mammoths, however, these animals cannot knock down trees to create grassland. Thus, Zimov and his colleagues have been using a tank to bring down the trees. One remarkable proposal then is to resurrect or recreate woolly mammoths to combat anthropogenic climate change.[7]

Genetic engineering uses ancient DNA and genome editing to resurrect or recreate a species. To do this, scientists must first reconstruct the genome of the extinct species. As noted, DNA can often survive much longer than cells after an organism dies. This is especially so with organisms that die in cold environments.[8] Full genomes have been reconstructed for mammoths, aurochs and passenger pigeons (*Ectopistes migratorius*).[9] The next step in resurrecting a species using genetic engineering is to determine what parts of the genome are responsible for which phenotypes. Next one takes the cell with the edited genome and creates a living organism. This often combines genetic engineering with SCNT in groups like mammals. However, genetic engineering currently cannot be used with SCNT in egg-laying species like birds or reptiles. One target for de-extinction using genetic engineering is the passenger pigeon. It is an awe-inspiring species, which existed in the billions. Aldo Leopold called it a "biological storm" (Leopold, 1949, p. 104). One recorded flock was 300 miles long and a mile wide. Martha, the last passenger pigeon, died in 1914 at the Cincinnati Zoo. The cause of the passenger pigeon's disappearance was predation by humans. For example, in 1878 hunters killed over 50,000 birds per day for 5 months straight.[10]

Currently, Ben Novak along with George Church working with others like Stewart Brand is planning to bring back the passenger pigeon.[11] The passenger pigeon's genome has already been sequenced and its closest living relative is the band-tailed pigeon (*Patagioenas fasciata*). They have already identified which parts of the band-tailed pigeons' genome that must be edited and replaced via CRISPR with the passenger pigeon's genes. Thereafter they

would grow band-tailed pigeon cells and edit the DNA within the nucleus with synthetic passenger pigeon DNA. As noted above, here complications arise since birds cannot be cloned. A fertilized ovum divides with its yolk moving down the oviduct, which is wrapped in albumen. When it reaches the isthmus, membranes are deposited around it and the eggshell is added. Currently, it is simply too difficult to enucleate and replace the nucleus of the egg without disrupting the process. Instead, primordial germ cell transplantation (PGCT) is used to create a chimera between the passenger and band-tailed pigeons. When PGCT is used where the donor and recipient are of distinct species, we have interspecies PGCT or iPGCT. From here, adult chimeras are mated together and scientists will work to establish a population of these birds.

A caveat of some importance is that these techniques for recreating or resurrecting species do not create an organism that is a copy of an extinct one. With back-breeding, at best an organism with the same phenotype of an extinct one is created. With genetic engineering, at best the same phenotype is generated from some of the same genes as the extinct organism, but many other parts of the genome are different (Sherkow and Greely, 2013). Even organisms that are cloned from extinct ones using SCNT are not copies since the mitochondria of the two will be different. Finally, there are different gene–environment interactions that occur with respect to the resurrected organisms and the extinct ones that can change their respective developments. However, many proponents of de-extinction suggest that creating exact copies of extinct species is not the primary aim. Beth Shapiro writes,

> In the majority of ongoing de-extinction projects, the goal is to create functional equivalents of species that once existed: ecological proxies that are capable of filling the extinct species' ecological niche. (Shapiro, 2017, p. 6)

Whether this is the aim of de-extinction research or not, can we nevertheless resurrect species?

## Is de-extinction possible?

Philosophers have argued about whether de-extinction is even possible. This issue is not whether it is difficult or even unlikely that a species will be resurrected or recreated. The concern is that it is physically or conceptually impossible.[12] On some views of what species are, they contend it is not possible (and possible on others) (Siipi and Finkelman, 2017; Finkelman, 2018). Let us consider one of the strongest arguments for this claim.

One popular view amongst philosophers of biology is that species are "historical entities" or, as it is sometimes put, they are individuals (Hull, 1978; Ereshefsky, 2000).[13] Consider the element gold. Something is gold if, and only if, it has atomic number 79. According to many philosophers and biologists, species are not like periodic elements. First, they are temporally bounded in that they have a beginning and an end in time. Second, they are spatially localized having a geographical distribution. Third, the organisms of species interact causally at a time (cohesion) and through time as well (integration) via various intraspecies processes. Gold is not like this since something is gold simply in virtue of having atomic number 79. It can occur in principle anywhere and elements of gold need not have anything to do with one another.

Here then is an argument against the possibility of de-extinction (for a discussion, see Campbell and Whittle, 2017; Slater and Clatterbuck, 2018). A species can evolve only if traits of organisms in the species are heritable. But a trait is heritable in a species only if organisms between generations are causally connected. Any entity whose parts are causally connected is a historical entity. Therefore, species are historical entities. De-extinction requires that organisms of a species exist at two separate times but which are not causally connected between those two times. But this is impossible. Hence, de-extinction is impossible.

To make this more concrete, let us consider the last generation of woolly mammoths on Wrangel Island and some that are created through cloning or genetic engineering (assuming it is successful). One might think that the former does not give birth to the latter since the latter was produced in a laboratory. So, they are not causally connected by heredity and thus the two populations do not share a history. They are not parts of the same historical entity, *Mammuthus primigenius.* But this argument fails (Campbell and Whittle, 2017; Slater and Clatterbuck, 2018; Campbell, 2022). To see why, consider a human baby who is born through in vitro fertilization. There is a clear sense in which ordinary reproduction has not occurred; it is a form of human-facilitated reproduction. Nevertheless, we do not think that the baby is not a member or part of *Homo sapiens.* They are nevertheless human. The same should be said regarding de-extinction. There is a causal connection between populations through human-facilitated reproduction as well.[14]

One might object that the woolly mammoth created through cloning or genetic engineering is a "mammophant" and not a genuine mammoth since it has Asian elephant DNA. It is a hybrid or a chimera. But hybridization and horizontal gene transfer occur throughout the natural world (Piotrowska, 2018). For example, all the brown bear (*Ursus arctos*) populations on the Admirality, Baranof and Chichagof islands possess polar bear mitochondria with less than 1% of their nuclear DNA from polar bear ancestry (Cahill et al., 2013). They are still brown bears. We know that humans for example interbred with Neanderthals and Denisovans, yet we are still members or parts of *H. sapiens* (Rogers et al., 2020). Additionally, mammophants would belong to a relevant woolly mammoth species according to some species concepts (Slater and Clatterbuck, 2018, pp. 8–11).

Another objection comes from Katz (2022). He claims that with biological systems, there is no design. Humans, and only humans, design objects. However, designing something even in part changes it into an artifact. By recreating and resurrecting species, we are creating artifacts that lack the "integrity" that natural objects possess. Additionally, this sort of design contributes to the domination of the natural world by humans which is morally wrong. There are several problems with this argument (for responses to Katz, see Browning and Veit, 2022; Lean, 2022; Preston, 2022; Reydon, 2022; Sandler et al., 2022; Turner, 2022). First, as many philosophers of biology and biologists argue, there can be design without designers (Kitcher, 1993). Natural selection is one such designing process.[15] Second, the argument rests on a flawed dualism between humans and the rest of nature. Humans are an extraordinarily unique and complicated mammal, but we are an evolved species amongst many others. As a species, we have radically changed our planet. Given most of us are moral agents, we can be morally responsible for those changes.[16] This concerns our status as *persons* and not as *humans.* Finally, even if we are dominating the planet and it is morally wrong, it is not clear how much a few instances of de-extinction will contribute to this. Many promote de-extinction as a form of restoration ecology; we are restoring ecological functions to ecosystems.[17]

A different sort of worry is that resurrected species are in some sense "inauthentic" (Siipi, 2014). When we say something is inauthentic, we are saying either that it is not identical to the object of interest or it does not share many of the properties that the target

of interest possesses (Siipi, 2014, pp. 77–78).[18] For example, there is no analog community for the woolly mammoths since many of the species that it would have existed with no longer exist. But the same can be said of translocation or assisted migration (Novak, 2018, p. 5). For example, wolves (*Canis lupus*) were reintroduced (or relocated) to Yellowstone National Park for the purposes of restoration and the recreation of trophic cascades (Fritts et al., 1997). Though the pack reintroduced did not descend from the last wolves eradicated in Yellowstone around 1926, they are of the same species. Likewise, as climate change alters the environments of many species, we may relocate them to places where their peer species are not present (Schwartz et al., 2012). This does not mean that they are a different or "inauthentic" species taxa any more than species aided with assistant migration.

In the end, there are a variety of technical issues associated with resurrecting or recreating species using de-extinction technologies. Nevertheless, it does appear possible to recreate or resurrect some lost species.

## Is de-extinction morally permissible?

We now consider the most important arguments for and against de-extinction (for surveys, see Sandler, 2014; Rohwer and Marris, 2018).

One common argument for de-extinction comes from considerations of restorative justice (Cohen, 2014; Jebari, 2016). When a moral agent harms a moral subject, the former owes the latter restitution. Humans harmed woolly mammoths for example by driving them extinct in combination with others factors like climate change (Martin, 2005). Thus, humans owe them restitution or compensation for that harm. The most apt form of restitution would be to resurrect the species. Therefore, we should resurrect woolly mammoths if it possible. As we saw in the last section, though there may be technical impediments to resurrecting the woolly mammoth, it is possible.

One immediate objection to the argument is the individual organisms harmed are long dead and *they* cannot be resurrected. This is true but beside the point since the harm of extinction befalls a species and not individual organisms.[19] A different objection to the argument from restoration is that *we* do not owe extinct species restitution since we did not drive them extinct. We were not alive then. We thus do not owe them anything (Campbell and Whittle, 2017).[20] This argument has force when we consider woolly mammoths, but it might have less force when we consider the passenger pigeon. After all, Americans killed passenger pigeons for their own material benefit. Thus, Americans who benefited from their extinction (even if they did not drive them extinct) *might* owe them restitution through de-extinction.[21]

Here is another argument for de-extinction from considerations of the conservation of biodiversity (Campbell and Whittle, 2017; Iacona et al., 2017). We should conserve and restore biodiversity. De-extinction is a means to conserve and restore species (and thus biodiversity). Thus, we should resurrect and restore species. Conservationists recognize the importance of preventing species from becoming threatened, endangered and eventually extinct. De-extinction raises the possibility of taking an extinct species and changing it conservation status (Campbell and Whittle, 2017, p. 91). Of course, our obligation to conserve and restore biodiversity is a prima facie obligation; it can be outweighed by other morally factors.[22] For example, if resurrecting a species creates a harmful invasive, then we may have good reason not to do it. As a special case of restoration (or rewilding), consider the restoration of the mammoth steppe (Josh Donlan et al., 2006; McCauley et al., 2017). By resurrecting or recreating the woolly mammoth, we restore ecological functions performed by megaherbivores in the recreation of grasslands that will help prevent climate change. Of course, this assumes that those functions can be performed in a changed environment, and there is not another appropriate species that could do it short of de-extinction.[23]

Let us turn to some of the most important objections to de-extinction.

The first argument against de-extinction concerns animal welfare (Kasperbauer, 2017; Browning, 2018; Browning and Veit, 2022). It is morally wrong to cause a sentient being unnecessary suffering. De-extinction will cause unnecessary suffering. Therefore, it is morally wrong to recreate and resurrect species. As a case in point, consider the bucardo cloned and born in 2003. Due to its deformed lungs, it lived for a brief time in tremendous pain (Cohen, 2014). De-extinction could lead to "miscarriage, stillbirth, early death, genetic abnormality and chronic disease" as the result of cloning (Browning, 2018, p. 789). And these are just the beginnings of the ethical issues since there are others regarding rearing and reintroducing these animals. However, conservation always involves a balance between protecting and restoring species and the welfare of individual organisms. For example, in places such as New Zealand endemic biodiversity is protected from predatory mammals through multi-kill traps and aerial poisons (Butler et al., 2014). Our obligations to conserve and restore biodiversity can be outweighed by conservations of animal welfare, but the extent to which there can be "compassionate conservation" is an active debate in conservation practice.

Another argument against de-extinction is the argument from hubris (or "playing God") (Minteer, 2014, 2019; Diehm, 2017). It is morally wrong to be hubristic (i.e., overly self-confident). De-extinction is hubristic insofar we overestimate our ability to predict and control resurrect or recreated species or ecosystems more generally. As Ben Minteer writes,

> Attempting to revive lost species is in many ways a refusal to accept our moral and technological limits in nature. De-extinction thus reflects a new kind of Promethean spirit that attempts to leverage our boundless cleverness and powerful tools for conservation rather than for human enhancement. But things did not end very well for Prometheus. (Minteer, 2014, p. 261)

There are several ways to respond to this argument. First, recreating or resurrecting species *can be* hubristic, but it is not necessarily so. For example, proponents of de-extinction are very much aware of the limitations of all the various techniques involved. As we discussed, we could only resurrect only a tiny fraction of all the species that have gone extinct. Second, this objection serves as an important moral call to continually avoid overestimating our abilities or underestimating the uncertainties or risks. Third, if sound, this argument would suggest we should avoid all manner of technologies not the least of which is de-extinction. After all, cloning and gene editing alone are only the most dazzling of our current technologies but conservationists also use camera traps, tracking tags, remote sensing, acoustic sensors, drones, eDNA and artificial intelligence. Should we avoid those as well because they may involve hubris?

A final ethical argument against de-extinction is that it is a poor allocation of conservation resources (Bennett et al., 2017). Conservation is an underfunded practice. Resources should be allocated to that which will do the most good in conserving and restoring

biodiversity. However, de-extinction will do little for that aim. Thus, funds should not be allocated to it. There are several objections that might be raised to this argument. First, there is a crucial assumption here: If money (or other resources) had not gone to a de-extinction project, then they would have gone to conservation work. But this may not be true since some de-extinction projects will be funded by those would not otherwise give to conservation. Efforts to bring back the heath hen in Martha's Vineyard appears to be such a case (Sandler, 2017, p. 2). Second, there is an assumption that ordinary conservation and de-extinction are mutually exclusive and that need not be the case. For example, there are good reasons to take tissue samples with DNA for "frozen zoos" or seed banks independent of whether they are used for de-extinction (Ryder et al., 2000). After all, if a species disappears "in the wild," conserving tissue samples may allow us to prevent them from going extinct.

We can now take stock of the moral arguments for and against de-extinction. The general conclusion we have found is that it is morally permissible to resurrect or recreate species except when there are outweighing morally relevant considerations. These moral considerations concern animal welfare, hubris and poor allocation of resources will sometimes outweigh obligations to recreate or resurrect species but not always. Thus, moral decisions regarding de-extinction will often need to be decided on a case-by-case basis.

## Conclusion

De-extinction is sometimes thought to be impossible given the nature of species and heredity. It may very well be unlikely, but it is does not appear impossible. Sometimes de-extinction is portrayed as *Jurassic Park* comes to life with all the morally failings in tow. But it is more accurately portrayed as a complicated moral issue without simple, easy answers.

**Open peer review.** To view the open peer review materials for this article, please visit http://doi.org/10.1017/ext.2023.4.

**Acknowledgements.** My thanks to Hank Greeley for putting together a conference in May 2013 entitled "De-extinction: Ethics, Law & Politics" at Stanford University. There were many different participants including Stewart Brand and Beth Shapiro. There were also other philosophers including Hilary Bok and Ronald Sandler. I learned a great deal from the experience, and it forms the background of this essay. Additionally, I thank the two anonymous reviewers for their very helpful comments.

## Notes

1. https://www.croptrust.org/work/svalbard-global-seed-vault/.
2. Christopher H. Lean argues that "De-extinction is better thought of as a set of techniques utilizing the remnants of extinct populations" (2020, p. 4). We need not recreate or resurrect species so much as use extinct populations as resources for introducing variation into taxa of interest. It is also worth noting that de-extinction techniques can be applied to several different units including genes, species, and ecosystems (Campbell and Whittle, 2017, pp. 8–11).
3. For another discussion of inclusive definitions of "de-extinction," see Campbell (2016).
4. There are limitations to back-breeding for the purposes of de-extinction. First, the ancestral trait must be found in an extant species which is closely related to the target extinct species. Second, the phenotypic matching needs to be the result of the same genes (or gene–environment interaction). Third, back-breeding can also create inbreeding depression lowering the fitness of individuals of the extant species, which may defeat the purpose of de-extinction.

5. In 2015, the auroch genome was fully sequenced from DNA extracted from a 6,750-year-old British auroch bone (Park et al., 2015). This opens the door to using some of the other de-extinction techniques we will discuss.
6. There are difficulties with this technique too. First, a small percentage of potential clones develop into living organisms. Second, cloning requires intact living cells from the extinct species which are often not available. Sometimes cloning can still occur when cells are not well-preserved. For example, an endangered subspecies of sheep, the mouflon (*Ovis gmelini*), was cloned from a nonviable cell of a dead sheep found in a field (Loi et al., 2001). However, for very recent extinct species, cells can be collected and preserved.
7. George Church has proposed resurrecting or recreating woolly mammoths through genetic engineering (https://www.techtimes.com/articles/226529/20180430/scientists-might-create-mammoth-elephant-hybrid-after-resurrecting-44-genes-will-start-with-mice-first.htm). The woolly mammoth's and the Asian elephant's genomes have been sequenced. Of the genetic differences between them, only 2,020 mutations affect genes that code for proteins. These genes affect phenotypic traits like hairiness, ear length, cold tolerance, and so forth. Using CRISPR gene editing technologies, Church and colleagues can excise the Asian elephant genes and replace them with woolly mammoth ones. Thus, they are in the first stages of creating what some call a "mammophant" – a cold-tolerant Asian elephant.
8. For example, a genome was reconstructed from a 700,000-year-old bone from a horse (Shapiro, 2017, p. 4).
9. There are limitations with this technique since DNA degrades more quickly in hot, wet environments than in cold, dry ones. Likewise, it is more difficult to reconstruct genomes when there are no close living relatives as with the New Zealand moa (*Dinornithiformes*).
10. https://www.si.edu/spotlight/passenger-pigeon.
11. Revive & Restore (https://reviverestore.org/about-us/) is an important organization dedicated to using biotechnology for conservation that was founded by Stuart Brand and Ryan Phelan. Several of their projects include recreating or resurrecting extinct species such as the passenger pigeon.
12. Philosophers have noted that there are different kinds of impossibility (and thus possibility). First, something is conceptually impossible it is logically inconsistent. Second, something is physically impossible when it is inconsistent with the laws of nature. Third, something is technologically impossible when it cannot occur given the current state of technology. The debates over de-extinction largely concern the first two notions of impossibility.
13. According to some philosophers, an individual is an entity that has a beginning and an end whose parts are integrated synchronically and diachronically (Mishler and Brandon, 1987). It is clear that a species' parts (i.e., organisms) may be only weakly integrated through processes like interbreeding even if they have a beginning via speciation and an end via extinction. Many philosophers as a result prefer the term "historical entity" to "individual."
14. It is worth emphasizing that even if resurrecting an extinct is possible, not every successful case of de-extinction is a resurrection. Back-breeding does not resurrect species so much as recreate them in our terminology. The argument here is some cases of de-extinction may be resurrections.
15. Interestingly, cultural evolution appears to be another such process (Boyd and Richerson, 1988; Richerson and Boyd, 2008). However, contrary to Katz (1997), human culture may design objects *without* intention.
16. A moral agent is a person who one who can make free decisions, can understand the consequences of their choices, and thus is morally responsible for their decisions. A moral subject is one who has a well-being; that is, can be benefited or harmed. Every moral agent is a moral subject but not necessarily vice versa.
17. To be fair, Katz (1997) opposes restoration ecology as well for similar reasons.
18. A different sort of worry is whether *authenticity* is even a scientific notion at all. For an attempt to show how it is a scientific notion, see Dudley (2012).
19. This raises ethical issues about whether species themselves can be harmed given that they are not sentient though their members or parts are (Campbell and Whittle, 2017; Kasperbauer, 2017). Due to space, we cannot address those issues here (though see Sandler, 2012).
20. This response turns on whether the wrongdoing was done by a group of humans or by our species.

21. For further discussion of this type of argument, see Cottrell et al. (2014) and Lean (2020).
22. An action is a prima facie obligation when it possesses a morally relevant feature and if that were the only morally relevant feature, then one would be obliged to do it.
23. One might object that the restoration of ecological function does not require us to resurrect or recreate a species. For example, a mammophant would be sufficient to restore the mammoth steppe. But this just is a form of de-extinction and so is not an objection to the above argument.

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
