## [Reviewer Report]

*Comments to Author*: This strikes me as a badly needed and (below issues notwithstanding) exceedingly clear review essay on the myriad debates about de-extinction. I would (generously) describe such debates as often exhibiting more heat than light, so it’s a boon to have a systematic, careful discussion of them. The author efficiently walks through many of the main critical arguments concerning the possibility and acceptability of de-extinction, deftly weaving together insights from biology, philosophy of biology, ethics, and environmental philosophy. I would, however, offer a few suggestions for them to consider in revising the paper. I direct the commentary now to the author.

As a minor terminological point, I don’t love the metaphor of ‘resurrection’. To my mind, it encourages a misleading view of what de-extinction requires. To resurrect a person (or non-human animal), whether by miraculous or technological, means bringing back the self-same being. Nothing like this is obviously required for de-extinction. Instead, and I think you agree with this interpretation, the idea is that we’re bringing back the same *kind* of organisms (or extending a lineage). I realize that conveying the former interpretation is not your intent and agree with you (and Shapiro) in finding ‘de-extincted’ to be an unappealing verb (perhaps ‘DE-ed’ for short is passable?), but I wanted to flag it. If you stick with ‘resurrection’, I would consider promoting the explanatory footnote to the main text and bear in mind throughout the potential for that metaphor to mislead (particularly on p. 4 in your discussion of Novak).

Concerning your discussion of Michael Hannah’s (2021) definition of extinction (as occurring “when the last individual belonging to that species dies”), I think a bit more clarity is needed on the question of whether this definition “trivializes the notion of extinction”. My take is that what is trivialized is the *impossibility* of de-extinction. This mirrors an argument of Delord (2014) that Clatterbuck and I discussed in our (2018, p. 3): if we did “bring back” members of a (thought-to-be-)extinct species, all we find is that this species wasn’t really extinct after all; for (to use Hannah’s definition) the last member of the species wasn’t dead. While this does trivialize the conceptual case against de-extinction, it doesn’t obviously trivialize the notion of *extinction*, which can remain a substantive question, though it *does* is arguably make our knowing whether an extinction has actually occurred — even when every individual of that species has died — effectively impossible, so long as we cannot rule out that further members won’t be produced at some time in the future. Knowing whether a species is extinct becomes (in part) a matter of prediction.

It seems to me that there is an ambiguity in our use of ‘last’ that applies to Hannah’s definition in a way that defangs the conceptual argument against de-extinction. Sometimes ‘last’ means “last in a given (potentially vaguely defined) time period”; other times it is not so restricted. For example, a relieved university dean says “Okay, that’s the last student to receive their diploma; commencement is now over.” It would be foolish to reply “No, it’s actually *not* over, as there are more students who will receive their diplomas in the future [e.g., next year]!” This sort of response is clearly to be distinguished from one in which the dean takes commencement to be over only to be alerted that they missed a name: what they *thought* was the last student due to receive a diploma wasn’t in fact the last (for the relevant time period).

Moving on: on p. 4, responding to Novak, you write that “[his] definitions of de-extinction are inclusive.” I’d explain in more detail what you mean by this.

p. 5: “Notice that back-breeding *recreates the auroch* by virtue of reestablishing auroch-like traits though the auroch is not resurrected.” Does it? I feel like the starred bit deserves a hedge or further argument (even setting aside the question of whether “recreating” a species constitutes a de-extinction).

p. 6: “Woolly mammoths are more closely related to Asian elephants (Elephas maximus) than African ones (Loxodonta). Thus, in principle they could be cloned with Asian elephant mothers.” I wouldn’t think the second sentence follows from the first. There may be independent reasons for thinking that the second sentence is true, of course (and the closer phylogenetic proximity might make it *more likely* that this is possible).

Considering the worry about Asian elephants being endangered (p. 6), it might be worth pointing out that there would presumably be other de-extinction candidates for which this is *not* the case. Reorganizing this discussion to start with the general point about closely-related surrogates and then raising the concern that some of these surrogates may entail their own ethical challenges could improve the flow of ideas here.

Also: It is an open-and-shut case that a “mammophant” wouldn’t be a genuine mammoth (supposing that it is a genetic clone of a mammoth)? Admittedly, there’s *some* reason for thinking not (the surrogacy issue, mitochondrial DNA, epigenetic effects, and so on), but if we ask the question to what species does this creature belong, on at least some species concepts, it seems plausible that the right answer is to the relevant mammoth species (see discussion in Slater & Clatterbuck 2018, pp. 8–11).

You begin section 2: “Philosophers have argued about whether de-extinction is even possible. This issue is not whether it is difficult or even unlikely that a species will be resurrected or recreated.” I would suggest clarifying the types of possibility you have in mind here — maybe ‘conceptually possible’ for the philosophical question and whether it is ‘technologically [or scientifically / practically] difficult / feasible’?

I’m not sure I’m grasping the authenticity argument (pp. 14–15); is it supposed to be relevant to the scientific or conceptual question of the possibility of de-extinction? If so, hard for me to see how it works; one obvious reply would be to point out that “authenticity” is not a precise concept of any biological science (that I’m aware of, anyway). This is compatible of course with your arguments, as I read them. Perhaps a very charitable reconstruction of the argument would be useful here.

When discussing criticisms of the moral case for particular de-extinction projects stemming from it not being *we* who “owe extinct species restitution since we did not drive them extinct” (since we weren’t alive in the Pleistocene!), you write that “This argument has force when we consider woolly mammoths, but it might have less force when we consider the passenger pigeon. After all, Americans killed passenger pigeons for their own material benefit.” Why not think it is simply equally bad, since, again, *we* were not the people who shot all the passenger pigeons (even if we’re compatriots with the culprits)! More could be said here I think. Is it that we — as Americans? as people living more proximately to the actual culprits? — somehow enjoy more advantaged from the extermination of passenger pigeons and so owe a debt of some sort? It’s not clear to me that a similar (more widely-applicable) advantage might not have stemmed from our Pleistocene forebears hunting mammoths. . . .

p. 19: in the discussion of the argument from hubris, I wonder if the response doesn’t strawman things a bit. Sure, the proponents are aware of the technical challenges and limitations and such, but I didn’t take the argument to be objecting to some kind of technical overconfidence, but to the very idea of intervening in this way at all — something that might eventually make us cavalier about letting species go extinct (or driving them there) in the first place. If that’s right, then the summary reply (“It would take a much stronger argument to show that by our nature we cannot avoid hubris”) would seem to miss the mark; the question is not whether this is something about “our nature” but whether it is an attitude that (whatever its source) should be resisted.

This strikes me as a badly needed and (below issues notwithstanding) quite clear review essay on the debates about de-extinction. I would (generously) describe these debates as exhibited more heat than light, so it’s a boon to have a systematic discussion of them. The author efficiently walks through many of the main critical arguments concerning the possibility and acceptability of de-extinction, deftly weaving together insights from biology, philosophy of biology, ethics, and environmental philosophy. I recommend acceptance, though I did have several suggestions that I’d encourage the author to consider while revising.

p. 2: I don’t love the metaphor of ‘resurrection’ since (to my mind) it seems carry a very particular (robust? weighty?) understanding of what de-extinction would require. I realize that this is not your intent (as is made clear in FN1), and I don’t have much better to suggest (other than ‘DE-ed’ perhaps), but I just wanted to flag it.

p. 3: It might be worth trying to be a little clearer on the discussion of whether ‘extinction’ is trivialized somehow by the response sketched. I know that there are arguments against de-extinction that trivialize *that* notion — e.g., DeLord argues (if I remember correctly) that if a member of a species is “brought back” then it was not truly extinct. Sounds like this discussion is in the neighborhood.

p. 4: “These definitions of de-extinction are inclusive.” Explain meaning of ‘inclusive’.

I might highlight the distinction between the technical senses of ‘resurrected’ and ‘recreated’ as you laid them out earlier (and what they amount to here) in the main text, since this is a key issue. On a fairly natural construal of ‘recreated’ (though *perhaps* not yours), recreating a species could entail resurrecting it.

“To see why, let us consider these three techniques….” (and following): This paragraph read awkwardly to me.

p. 5: “Notice that back-breeding **recreates the auroch** by virtue of reestablishing auroch-like traits though the auroch is not resurrected.” Does it? I feel like the starred bit needs a hedge.

p. 6: “Woolly mammoths are more closely related to Asian elephants (Elephas maximus) than African ones (Loxodonta). Thus, in principle they could be cloned with Asian elephant mothers.” I wouldn’t think the second sentence follows from the first. . . .

Considering the worry about Asian elephants being endangered, it might be worth pointing out that there would presumably be other candidates for which this is *not* the case. I might consider reorganizing this discussion to start with the general point about closely-related surrogates and then raise the concern that some of these surrogates may involve their own ethical challenges.

p. 7: “However, these animals unlike woolly mammoths, cannot knock down trees to create grassland.” Awkward. Rephrase as: “Unlike woolly mammoths, however, these animals cannot knock down trees to create grassland.

In the footnote (“Thus, there are…”)  ‘…they are’.

Also: It is an open-and-shut case that a “mammophant” wouldn’t be a genuine mammoth (supposing that it is a genetic clone of a mammoth)? Admittedly, there’s *some* reason for thinking not (the surrogacy issue, mitochondrial DNA, epigenetic effects, and so on), but if we ask the question to what species does this creature belong, on at least some species concepts, it seems plausible that the right answer is to the relevant mammoth species.

p. 10: the Shapiro quote is a little strange. Why think that there’s a single goal? That seems like it fits in well to your discussion here and could be brought out a bit more clearly. Again, what we’re talking about under the heading of ‘resurrection’ obviously matters a great deal.

“Philosophers have argued about whether de-extinction is even possible. This issue is not whether it is difficult or even unlikely that a species will be resurrected or recreated.” Clarify the types of possibility here — maybe ‘conceptually possible’ for the philosophical question and whether it is ‘technologically [or scientifically / practically] difficult’?

p. 15: the authenticity argument seems to me very odd. One obvious reply to my mind is to point out that authenticity is not a precise concept of any biological science (that I’m aware of, anyway). This is compatible of course with your arguments; thought it might be worth mentioning.

p. 16: “This argument has force when we consider woolly mammoths, but it might have less force when we consider the passenger pigeon. After all, Americans killed passenger pigeons for their own material benefit.” Less force, yes, but why not think it is equally bad, since again *we* (even if we’re both Americans) were not the people who shot all the passenger pigeons! More could be said here I think. Is it that we (as Americans? as people living more proximately to the actual culprits?) somehow enjoy more advantaged from the extermination of passenger pigeons and so owe a debt? It’s not clear to me that a similar (more widely-applicable) advantage might not have stemmed from our Pleistocene forebears hunting mammoths. . . .

p. 19: in the discussion of the argument from hubris, I wonder if the response doesn’t strawman things; sure, the proponents are aware of the technical challenges and limitations and such, but I didn’t take the argument to be objecting to some kind of technical overconfidence, but to the very idea of intervening in this way at all — something that might eventually make us cavalier about letting species go extinct (or driving them there) in the first place. If that’s right, then the summary reply (“It would take a much stronger argument to show that by our nature we cannot avoid hubris”) would seem to miss the mark; the question is not whether this is something about “our nature” but whether it is an attitude that (whatever its source) should be resisted.

“After all, cloning and gene editing **along** are…”  ‘alone’ I assume.

p. 20: “there are good reasons to take tissue samples with DNA for “frozen zoos” or seed banks independent of whether they are used for de-extinction….” Briefly explain / mention what these good reasons are?

I found the paragraph right before the conclusion to read a bit awkwardly. Ditto for the conclusion (seemed to be rapidly composed, with some grammatical issues). Why not take a more substantive, thematic stock of the territory covered?

---

## [Reviewer Report]

*Comments to Author*: This looks good to go. I have some minor citations and comments that could be incorporated but this looks like a good summary of this literature.

Page 4. On inclusive definitions. Campbell has an interesting variation on an inclusive definition where it is where it is the expression of adaptive traits “for de-extinction

purposes, Pb counts as being the same species as Pa just to the degree that many of

the evolutionarily adaptive traits possessed by the members of Pa have been genetically inherited by and are phenotypically expressed by members of Pb”

Page 12. The argument at the end of the page may have some similarities with: Piotrowska, M. (2018). Meet the new mammoth, same as the old? Resurrecting the Mammuthus primigenius. Biology & Philosophy, 33(1), 1-16.

Page 15. I think that the following paper deserves a citation for their engagement with the duties argument also: Cottrell, S., Jensen, J. L., & Peck, S. L. (2014). Resuscitation and resurrection: The ethics of cloning cheetahs, mammoths, and Neanderthals. Life Sciences, Society and Policy, 10(1), 3

Page 16. Lean 2020 replies to the duty’s argument at length.

---

## [Editor Report]

*Comments to Author*: Dear Dr. Odenbaugh,

I now have two very positive reviews of your manuscript. While both reviewers recommend acceptance of your paper, they do both have suggestions for improving the manuscript. Therefore, I am recommending minor revision to give you time to incorporate the suggestions you find useful. However, I do not anticipate sending it out for review again. I look forward to seeing a revised manuscript along with a detailed cover letter responding to the reviewer comments. 

Sincerely,

Kate Lyons

Senior Editor

---

## [Editor Report]

*Comments to Author*: Dear Dr. Odenbaugh,

Thank you for revising your manuscript to incorporate the suggestions of the two reviewers. I am happy to recommend it for publication. 

Best wishes,

Kate Lyons